# Mechanistic Aspects for the Modulation of Enzyme Reactions on the DNA Scaffold

**DOI:** 10.3390/molecules27196309

**Published:** 2022-09-24

**Authors:** Peng Lin, Hui Yang, Eiji Nakata, Takashi Morii

**Affiliations:** Institute of Advanced Energy, Kyoto University, Kyoto 611-0011, Japan

**Keywords:** DNA scaffold, enzyme reaction, catalytic enhancement, cascade efficiency, mechanism

## Abstract

Cells have developed intelligent systems to implement the complex and efficient enzyme cascade reactions via the strategies of organelles, bacterial microcompartments and enzyme complexes. The scaffolds such as the membrane or protein in the cell are believed to assist the co-localization of enzymes and enhance the enzymatic reactions. Inspired by nature, enzymes have been located on a wide variety of carriers, among which DNA scaffolds attract great interest for their programmability and addressability. Integrating these properties with the versatile DNA–protein conjugation methods enables the spatial arrangement of enzymes on the DNA scaffold with precise control over the interenzyme distance and enzyme stoichiometry. In this review, we survey the reactions of a single type of enzyme on the DNA scaffold and discuss the proposed mechanisms for the catalytic enhancement of DNA-scaffolded enzymes. We also review the current progress of enzyme cascade reactions on the DNA scaffold and discuss the factors enhancing the enzyme cascade reaction efficiency. This review highlights the mechanistic aspects for the modulation of enzymatic reactions on the DNA scaffold.

## 1. Introduction

Living organisms have evolved over millions of years to build a complex metabolic network containing thousands of enzymatic reactions for their survival [1]. Enzymes are spatially organized in the cell to implement specific sequential reactions via the strategies of compartmentalization [2]. The spatial organization of enzymes often relies on the specific scaffolds of proteins or the membrane to achieve the high efficiency and the specificity of enzymatic reactions [3]. Ribulose 1,5-bisphosphate carboxylase/oxygenase (RuBisCO) and carbonic anhydrase packed in the protein shell of carboxysome [4] and cytochrome P450 enzymes anchored on the membrane of endoplasmic reticulum [5] are the typical examples. In these compartments, the reactants in low concentration are believed to be effectively transferred through spatially arranged enzymes, thereby channeling metabolites to drive favorable reactions and preventing the toxic side reactions by intermediates [6]. It is our challenge to understand the efficient bioenergetic processes of nature and to construct human-engineered energy utilization systems [7].

To mimic and understand the natural systems, a wide variety of carriers has been built for the attachment of enzymes, as has been reviewed previously [8,9,10,11]. The further applications of the carriers such as protein, liposome or polymersome are challenged by the structural programmability of carriers and the spatial arrangement of enzymes [12]. These obstacles are tackled by means of DNA nanotechnology [13]. A typical example of DNA nanostructures, DNA origami, folds a long, single-stranded DNA into the predesigned, two-dimensional (2D) and three-dimensional (3D) DNA scaffolds with accurate addressability, providing the ideal templates for the assembly of enzymes [14,15]. Individual or multiple enzymes have been spatially arranged on the DNA scaffolds with precise control over the enzyme orientations, interenzyme distance and the stoichiometry of enzymes [16].

Interestingly, the catalytic enhancement of single type of enzyme assembled on the DNA scaffolds has been widely observed [17]. This phenomenon has been attributed to the substrate affinity to the negatively charged DNA scaffold surface by electrostatic interactions, lower local pH on the DNA scaffold surface, reduced adsorption of scaffolded enzymes on the reaction vessels or the ordered hydration layer attracted by the DNA surface [18]. However, the notion of whether these proposed mechanisms can apply in general for the catalytic enhancement of DNA-scaffolded enzymes remains controversial. Besides the single type of enzyme, the cascade reactions of multi-enzyme assembled on the DNA scaffold were also extensively studied [19,20]. While the close interenzyme distance, optimal spatial organization of enzymes and a confined DNA environment have been proposed to be the main factors enhancing the efficiency of enzyme cascade reactions on the DNA scaffold, the actual mechanisms remain to be elucidated [21].

In this review, we first describe the characteristics of spatial organization of enzymes in cells to highlight the elegance of natural strategies and the importance of these strategies on the metabolic processes. Then we review the reactions of a single type of enzyme assembled on the DNA scaffold and discuss the previously proposed mechanisms for the catalytic enhancement of DNA scaffolded enzymes. We also summarize the current progress of the cascade reactions of the multi-enzyme assembled on the DNA scaffold and discuss the factors that can enhance the cascade efficiency. While there are many reviews describing the application of DNA scaffolds for enzyme reactions [21,22,23,24], this review highlights the mechanistic aspects for the modulation of enzymatic reactions on the DNA scaffold. Understanding the mechanisms behind the enhanced catalytic activity or cascade efficiency of these DNA scaffolded individual- or multi-enzyme will accelerate the wide applications of artificial metabolic systems.

## 2. Construction of the Artificial Enzymatic Reaction Systems Inspired by Nature

To implement the efficient enzyme cascade reactions for the biochemical transformation, cells utilize the strategy of compartmentalization by forming the membrane-bound organelles (e.g., mitochondria, chloroplasts and peroxisome), bacterial microcompartments and multi-enzyme complexes [25]. The spatial organization of enzymes on the specific scaffolds (e.g., proteins or membrane) exerts the high efficiency and specificity of cascade reactions by increasing the concentration of reactants, reducing the toxic intermediates and competing reactions and overcoming the unfavorable enzyme kinetics [26]. To mimic the natural systems, a wide ranges of materials have been applied to construct the artificial scaffolds for enzyme reactions in vitro [18].

### 2.1. The Spatial Organization of Enzymes in Nature 

Substrate channeling is the transportation of the intermediates from one enzyme active site to the next without the release into solution [26]. The typical example of substrate channeling is tryptophan synthase. This α_2_β_2_ tetrameric enzyme complex catalyzes a two-step cascade reaction that converts indole-3-glycerol-phosphate and serine to tryptophan. As the intermediate, indole is directly transferred through the 25 Å long hydrophobic tunnel bridging the two active sites (Figure 1a) [27]. Such hydrophobic tunnel is also found in carbamoyl–phosphate synthetase that channels carbamates to the neighboring active site. The substrate channeling couples the first hydrolysis reaction of glutamine and the successive reactions to maintain the proper stoichiometry despite of the three-order-of magnitude difference in *K*_m_ for NH_3_ in the carbamate synthesis and that for glutamine in the first reaction [28]. Instead of using hydrophobic tunnel, the malate dehydrogenase–citrate synthase (MDH–CS) complex utilizes the electrostatic guidance to channel the metabolite oxaloacetate to achieve a high flux though the MDH–CS pair by overcoming the unfavorable kinetics of MDH forward reaction in the citric acid cycle [29].

Metabolons feature the dynamic assembly and disassembly of the enzyme complexes [30]. These highly transient protein assemblies are believed to allow the direct channeling of the intermediates from one enzyme to the next consecutive enzyme in the metabolic pathway. Purinosome [31], dhurrin metabolon [32] and glucosome [33] are the typical examples. While the spatiotemporal enzyme assemblies are proposed to enhance the metabolic flux and biochemical transformation, the study of the spatial organization of metabolon remains challenging due to the transient and complex interactions of proteins involved [30]. In 2008, the human purinosome was first identified, in which the ten chemical steps of de novo purine biosynthesis (DNPB) were catalyzed by six enzymes. Two human enzymes involved in DNPB were fused to either a green fluorescent protein (GFP) or an orange fluorescent protein (OFP) and transiently expressed in the purine-rich or purine-depleted cells. The cytoplasmic clusters formed by these two enzymes were observed by fluorescence microscopy in purine-depleted cells. This provided the evidence for the formation of the enzyme complex “purinosome” [34]. Afterward, the protein–protein interactions of purinosome and the regulation of its formation were extensively studied [35]. In 2020, Pareek et al. [36] used metabolomics and in situ three-dimensional sub-micrometer chemical imaging of single cells by gas cluster ion beam secondary ion mass spectrometry (GCIB-SIMS) to directly visualize DNPB by purinosome. It was proposed that purinosome consisted of at least nine enzymes that functioned synergistically to increase the pathway flux. Moreover, these DNPB metabolons were hypothesized to locate proximal to the mitochondria. In a recent study, Pedley et al. [37] found the heat shock protein 90 kDa (Hsp90) would help to regulate the physical properties of the purinosome and maintain the liquid-like state inside HeLa cells (Figure 1b). This finding provides novel insights into how the liquid–liquid phase separation drives the formation of metabolon in the human cell.

As the best-studied bacterial microcompartments, carboxysomes play a central role in the carbon concentrating mechanism [4]. Two enzymes, RuBisCO and carbonic anhydrase (CA), are packed in the polyhedral protein shells of carboxysome (Figure 1c). CA converts bicarbonate (HCO_3_^−^) to CO_2_, which is subsequently consumed by RuBisCO in the carbon fixing reaction. The diffusion of negatively charged HCO_3_^−^ into the protein shell is promoted by the positively charged pores of carboxysome, while the concentration of uncharged CO_2_ is increased in the compartment, resulting in the enhanced reactivity of RuBisCO [4]. Moreover, RuBisCO molecules have been hypothesized to form the condensate by the liquid–liquid phase separation, which provides the understanding of carboxysome biogenesis. The function of the protein shell of carboxysome in RuBisCO condensation remains to be elucidated [38].

Membrane plays important roles in many biological processes in cells. Cyanobacterial thylakoid membrane possesses a unique structure carrying both photosynthetic and respiratory electrons transfer chains, allowing the performance of both oxygenic photosynthesis and aerobic respiration in the same cellular compartment (Figure 1d) [39]. The membrane of endoplasmic reticulum or mitochondria provides the scaffold for anchoring the cytochromes P450, an important class of enzymes involved in the biotransformation of many endo- and exogenous compounds. The structure and dynamics of cytochrome P450 on biomembranes have been reviewed [5]. While the scaffolds in nature are proposed to assist the co-localization of enzymes, maintain the enzyme functions and enhance the enzymatic reactions, the actual working mechanisms of scaffolds in the natural compartments remain unclear.

**Figure 1 molecules-27-06309-f001:**
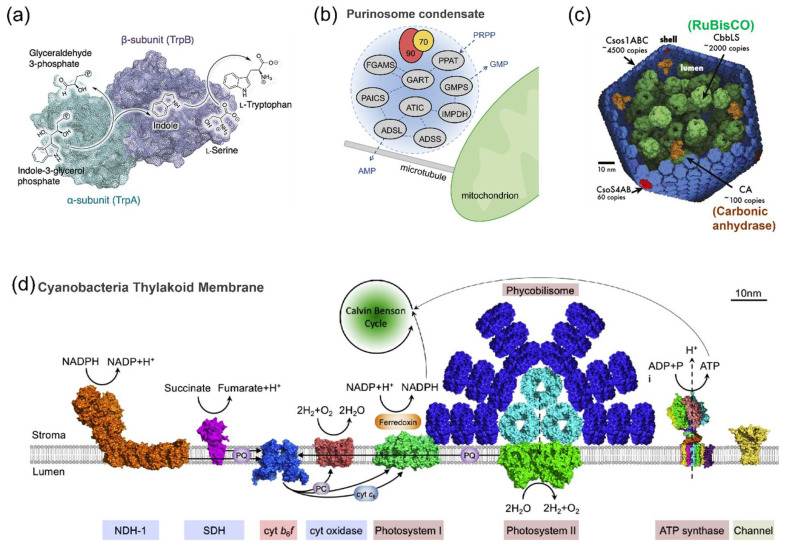
Spatial organization of enzymes in nature systems. (**a**) The structure of tryptophan synthase that converts indole-3-glycerol-phosphate and serine to tryptophan. This α_2_β_2_ tetrameric enzyme complex is formed by two subunits: α-unit (TrpA) and β-unit (TrpB) [27]. (**b**) The formation of purinosome condensate for de novo purine biosynthesis. The purinosome condensates are proposed to conduct the channeled conversion of phosphoribosyl pyrophosphate (PRPP) to guanosine 5′-monophosphate (GMP) or adenosine 5′-monophosphate (AMP). Hsp70/Hsp90 chaperone machinery is involved in the assembly of the purinosome [37]. (**c**) An illustration of carboxysome with RuBisCO and carbonic anhydrase packed inside the protein shell, scale bar: 2 µm [4]. (**d**) Schematic model of cyanobacterial thylakoid membrane carrying the photosynthetic and respiratory electron transport complexes [39]. Reprinted with permission from: (**a**) Ref. [27]. Copyright 2017 American Chemical Society; (**b**) Ref. [37]. Copyright 2022 Elsevier Ltd.; (**d**) Ref. [39]. Copyright 2016 Elsevier Ltd.

### 2.2. Enzymatic Reactions on Various Carriers

The reactions of individual or multienzymes have been conducted on a wide range of carriers, such as metal–organic frameworks (MOFs) [40], hydrogel [41], graphene oxide [42], liposome [43], polymersome [44], proteins [45] and DNA nanostructures [46]. These artificial scaffolds are suggested to enhance the enzyme stability, reusability or catalytic ability, expanding the applications of enzyme in different fields such as the biosynthesis of value-added chemicals. These scaffolds are also applied to mimic and understand the natural systems such as complex metabolic pathways or molecular transport.

Yoshimoto et al. [47] entrapped glucose oxidase (GOx) and catalase into the liposome membrane (Figure 2a). The hydrogen peroxide (H_2_O_2_) produced by GOx in the glucose oxidation reaction inside the liposome was decomposed by catalase. A remarkable protection effect of the liposome membrane on catalase activity inside the liposome was observed. The presence of outer membrane protein F (OmpF) enhanced the transport of glucose molecules from the exterior of the liposomes to the interior and increased the enzyme activity of GOx by 17 times compared with that of GOx encapsulated in the liposome in the absence of OmpF. This system demonstrates the typical example of the application of liposome as an enzyme carrier. Klermund et al. [44] compartmentalized the three-step reaction in a polymersome to synthesize CMP-*N*-acetylneuraminic acid (CMP-Neu5Ac) (Figure 2b). *N*-Acetylneuraminate lyase (NAL) and CMP-sialic acid synthetase (CSS) were inserted into polymersome, while *N*-Acyl-D-glucosamine 2-epimerase (AGE) with allosteric activator ATP was encapsulated inside the polymersome. Channel proteins OmpF enabled a selective mass transport across the polymer membrane. The incorporation of OmpF into the membrane restrained the cross-inhibitions in enzyme cascade reactions. The overall throughput was enhanced 2.2-fold compared to the reaction by free enzymes.

Protein scaffolds are also widely applied as the templates for enzymatic reactions. Zhang et al. [45] utilized the shell proteins from the ethanolamine utilization bacterial microcompartment (EutM) to attach enzymes by SpyTag–SpyCatcher conjugation strategy. Alcohol dehydrogenase (ADH) and amine-dehydrogenase (AmDH) were co-immobilized on the protein scaffold to convert alcohols to chiral amines in a highly enantioselective manner (Figure 2c). To construct an artificial carboxysome, Raphael et al. [48] used an electrostatic tagging system to co-encapsulate RuBisCO and carbonic anhydrase (CA) in an engineered protein cage based on lumazine synthase from *Aquifex aeolicus*, AaLS-13 (Figure 2d). RuBisCO and CA were genetically fused to the positively supercharged variants of green and yellow fluoresce proteins, GFP(+36) and TOP(+36), to obtain G-RuBisCO and T-CA, respectively. These positively charged constructs were encapsulated by the AaLS-13 capsids containing a negatively charged lumenal surface. While a significant kinetic effect of co-entrapped CA on the enzyme activity of RuBisCO was not observed under ambient or oxygen saturated conditions, this system nonetheless provided the new strategy of constructing the artificial organelles.

**Figure 2 molecules-27-06309-f002:**
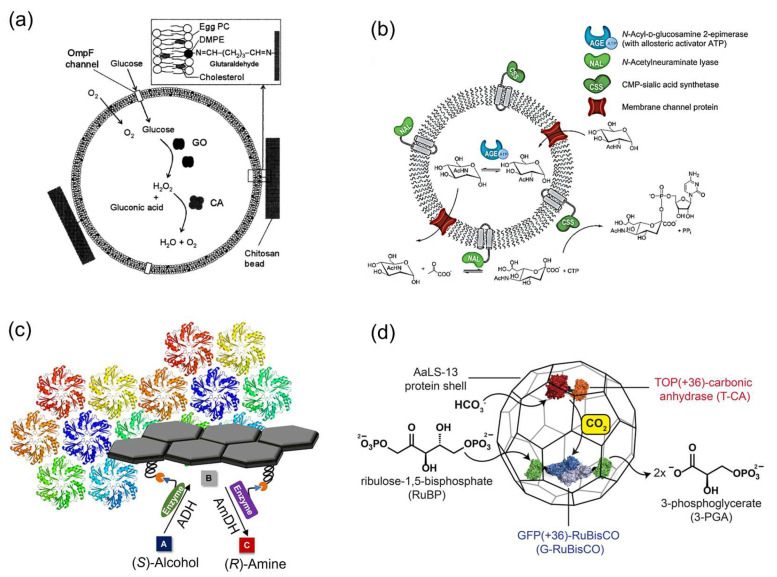
Enzymatic reactions on various carriers. (**a**) Schematic drawing of the liposome system containing entrapped glucose oxidase (GOx), catalase and membrane-embedded OmpF, which was immobilized into chitosan beads [47]. (**b**) Enzyme cascade reaction compartmentalized in a polymersome for the synthesis of CMP-*N-*acetylneuraminic acid (CMP-Neu5Ac). The *N*-Acyl-D-glucosamine 2-epimerase (AGE) is separated from the incompatible component CTP by encapsulation in polymersomes. The *N*-Acetylneuraminate lyase (NAL) and the CMP-sialic acid synthetase (CSS) are immobilized on the surface to form a single biocatalytic entity [44]. (**c**) Co-immobilization of alcohol dehydrogenase (ADH) and amine-dehydrogenase (AmDH) on the protein scaffolds for the synthesis of chiral amine [45]. (**d**) Schematic representation of an artificial carboxysome constructed by protein shell based on lumazine synthase from *Aquifex aeolicus*, AaLS-13. RuBisCO and carbonic anhydrase (CA) were genetically fused to the positively supercharged variants of green and yellow fluoresce proteins, GFP(+36) and TOP(+36), to obtain the constructs G-RuBisCO and T-CA, respectively [48]. Reprinted with permission from: (**a**) Ref. [47]. Copyright 2005 John Wiley and Sons; (**b**) Ref. [44]. Copyright 2017 American Chemical Society; (**c**) Ref. [45]. Copyright 2018 American Chemical Society; (**d**) Ref. [48]. Copyright 2016 American Chemical Society.

## 3. Catalytic Enhancement of Single Type of Enzyme Assembled on the DNA Scaffold

While the soft materials show their own advantages and the potential as the enzyme scaffolds, the further applications of carriers such as liposome, polymersome or protein face the difficulties in controlling the enzyme loading positions and stoichiometry. Therefore, the scaffolds that can overcome these challenges are required for the enzyme assembly in vitro. Given the predominant advantages of structural programmability and accurate addressability, DNA nanostructures are considered as the ideal platforms for the assembly of enzymes [49]. In this section, we review the reactions of a single type of enzyme assembled on the DNA scaffolds and discuss the previously proposed mechanisms for the catalytic enhancement of enzyme by the DNA microenvironment. Understanding the origins of enhanced activity of DNA-scaffolded enzymes will expand their practical applications.

### 3.1. DNA Origami Scaffold

The past two decades have witnessed the rapid development of DNA origami and its applications [50]. In 2006, Rothemund [14] created DNA origami that folds a long, circular, single-stranded DNA template (7-kilobase) into desired two-dimensional (2D) shapes with the aid of over 200 short oligonucleotides (staple strands). Nonperiodic structures, such as square, rectangle, star and smiley face, were obtained (Figure 3a). Featuring preparation simplicity, structural programmability and high folding yield, DNA origami plays an important role in the development of structural DNA nanotechnology. In 2009, Douglas et al. [15] extended the DNA origami method to build custom three-dimensional structures formed as pleated layers of DNA helices in the honeycomb lattices, providing a general route to the construction of complex 3D DNA nanostructures (Figure 3b). By using the computer-aided design software for DNA origami nanostructures such as caDNAno, the staple sequences for folding newly designed DNA nanostructures are easily generated [51,52].

With the development of the DNA origami technique, a number of DNA nanostructures with complexity were constructed, such as DNA box, DNA nanoflask and hollow DNA sphere (Figure 3c) [53,54,55]. Featuring tremendous self-assembly properties and addressability, DNA scaffolds provide the ideal platforms for the assembly of functional macromolecules such as proteins. To locate proteins of interest on DNA scaffolds, various strategies of DNA–protein conjugation have been developed with noncovalent and covalent conjugations [22]. Besides the static DNA origami structures, the dynamic DNA nanotechnology enables the construction of various reconfigurable DNA origami structures induced by the hybridization of short DNA, aptamer–ligand interactions, temperature, pH, ion and electric field with controlled translational, rotational or more sophisticated movement [56,57,58,59,60,61,62]. These dynamic DNA structures have been widely applied in the fields of drug delivery, diagnosis, biosensing and biocatalysis, which has been reviewed previously [63,64,65,66,67].

**Figure 3 molecules-27-06309-f003:**
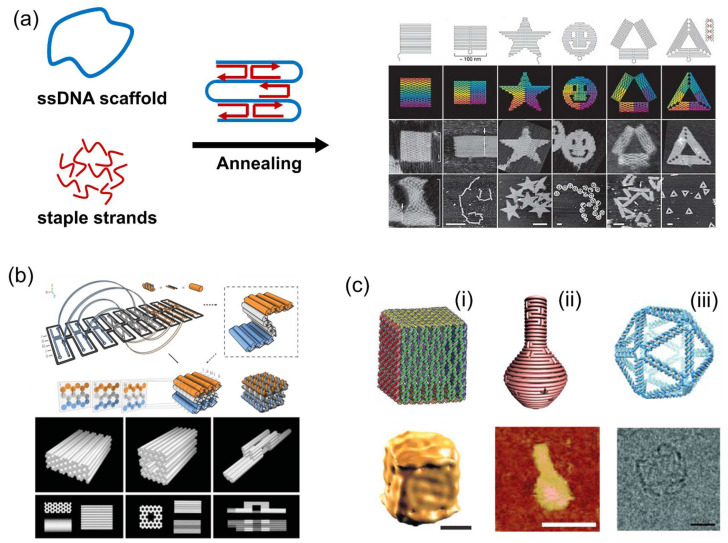
DNA origami technique and DNA origami scaffolds. (**a**) DNA origami using a circular single-stranded DNA template (7 kb) that folds into predesigned shapes with the aid of DNA staple strands [14]. (**b**) Design and folding of three-dimensional DNA origami structures [15]. (**c**) Examples of three-dimensional DNA origami structures. (**i**) A DNA origami box ant its Cryo-EM image, scale bar: 20 nm [53]. (**ii**) A DNA nanoflask and its AFM image, scale bar: 75 nm [54]. (**iii**) A 3D DNA spherical structure and its TEM image, scale bar: 10 nm [55]. Reprinted with permission from: (**a**) Ref. [14]. Copyright 2006 Springer Nature; (**b**) Ref. [15]. Copyright 2009 Springer Nature; (**c-i**) Ref. [53]. Copyright 2009 Springer Nature; (**c-ii**) Ref. [54]. Copyright 2011 The American Association for the Advancement of Science (AAAS); (**c-iii**) Ref. [55]. Copyright 2016 AAAS.

### 3.2. “Favorable Microenvironment” Provided by DNA

The enzyme scaffolded by DNA structures often displays enhanced activity and stability over its free form; however, the actual mechanisms for the higher catalytic ability are still under debate [68]. As studied previously, Glettenberg et al. [69] covalently conjugated peroxidase to different DNA oligonucleotides (ODN). The ODN markedly influenced the catalytic of tethered-enzyme in a DNA sequence-dependent manner. This phenomenon was attributed to the interactions such as hydrogen bonding or electrostatic contacts between ODN and the heme-containing catalyst. Rudiuk et al. [70] conjugated β-lactamase with a branched DNA complex constructed by four λ DNA (48.5 kbp), which underwent a dramatic and reversible higher-order structural transition regulated by spermine (SPM^4+^) and NaCl. In this study, the enhanced catalytic activity of enzyme was attributed to the “favorable microenvironment” composed of the giant and ordered DNA molecules. Then an interesting and important question arises: What is “favorable DNA microenvironment”, and what is the chemistry behind it? 

### 3.3. Protection Effect Derived from DNA Scaffold

Featuring the characteristics of a highly negatively charged surface, DNA scaffolds have been reported to protect enzymes from deactivation. Timm et al. [71] tethered the *S*-selective NADP^+^/NADPH-dependent oxidoreductase Gre2 from *S. Cerevisiae* on rectangular DNA origami structures through chemo- and site-selective protein–DNA coupling methods. Gre2 was first fused with the Halo-tag or SNAP-tag to obtain Halo-Gre2 or SNAP-Gre2. Chlorohexane (CH) or benzylguanine (BG) was then incorporated as suicide ligand into the DNA origami scaffold to facilitate the crosslinking reaction of the Halo-Gre2 or SNAP-Gre2. To assess the enzyme stability, the activity of the free enzyme or DNA-scaffolded enzyme was measured immediately after preparation (incubation for 0 h) or after incubation for 2.5 h. A significantly increased activity of the DNA origami-tethered enzymes as compared to free enzymes in solution was observed (Figure 4). Considering the phenomenon that the free enzymes lost significant activity upon incubation for 2.5 h, it was suggested that the large, highly charged DNA nanostructures would protect the enzyme against denaturation. In particular, the adsorption of scaffolded enzymes on the surface of the reaction vessels would be reduced. Moreover, DNA structures have been reported to protect the encapsulated protein from the hydrolysis catalyzed protease [72].

### 3.4. Substrate Affinity to the DNA Scaffold

DNA scaffolds with the high-density negative charge have been proposed to attract molecules through electrostatic interactions. Lin et al. [73] assembled horseradish peroxidase (HRP) on a triangular DNA scaffold to study the substrate–scaffold interactions by using various substrates. *p*-Aminophenol (AP) with a positive charge was first used as a substrate for the oxidation reaction catalyzed by HRP. An over 250% increase in the activity of scaffolded HRP over the free enzyme was observed with AP. This enhancement was decreased to 153 ± 21, 131 ± 11 and 158 ± 24% in the presence of 50, 150 and 300 mM NaCl, respectively, suggesting that the electrostatic interaction between the positively charged AP and the negatively charged DNA scaffold partly contributed to the catalytic enhancement. In addition, the reactions of scaffolded HRP with phenol (P), *p*-hydroxybenzoic acid (HBA), *o*-phenylenediamine (OPD) and 3,3′,5,5′-tetramethylbenzidine (TMB) showed the kinetic enhancement of 244 ± 6%, 197 ± 6%, 130 ± 16% and 138 ± 12%, respectively. The reduced *K*_m_ values of scaffolded HRP for these substrates supported the increased affinity to the substrates. The activity of the scaffolded HRP with 2,2′-azino-bis(3-ethylbenzothiazoline-6-sulphonic acid) (ABTS) was lower than the free HRP due to the negative charge of ABTS. Interestingly, the plot of the kinetic enhancement of scaffolded HRP against the binding energy of substrate to HRP suggested that the DNA scaffold enhanced HRP activity following the Sabatier Principle, a “just and right” manner (Figure 5). In a recent study, Kosinski et al. [74] encaged thrombin, a model of allosterically regulated serine proteases, into the cavities of DNA scaffolds with distinct structural and electrostatic characteristics. The hydrolysis reactions with peptide substrates carrying different charge indicated that the reaction rates were affected by DNA/substrate electrostatic interactions. These examples indicate the effect of DNA–substrate interactions on the enzymatic reactions.

### 3.5. Ordered Hydration Layer on the DNA Scaffold Surface

Zhao et al. [75] have observed 3- to 10-fold activity enhancements of five different enzymes, GOx, HRP, glucose-6-phosphate dehydrogenase (G6pDH), malic dehydrogenase (MDH) and lactic dehydrogenase (LDH), individually encapsulated in DNA cages. It was hypothesized that enzymes were stabilized by the highly ordered, hydrogen-bonded water environment formed by the negatively charged DNA cage surface, in which the stabilization of the hydrophobic interactions of a folded protein was induced by the solvent entropy penalty upon protein unfolding. The activity of encapsulated G6pDH reduced to approx. 25% activity in the presence of 1 M NaCl. This was suggested that Na^+^ would shield the negative charge on the DNA surface and disrupt the surface-bound hydration layer. However, the high concentration of salts containing the cations such as Na^+^, K^+^ and NH_4_^+^ also strongly inhibited the activity of free G6pDH. The DNA hydration layer may play an important role in the modulation of enzyme reactions on the DNA scaffold, but the stabilization of proteins may not be the general mechanism for enhancing the activity of the DNA-scaffolded enzymes.

### 3.6. Local pH Environment

Besides the above factors, local pH environment near the enzyme was proposed as a critical factor to improve the enzyme activity on DNA scaffold. Zhang et al. [76] suggested that activity enhancement of enzymes (GOx or HRP) located on DNA scaffolds derived from the lower pH on the negatively charged DNA scaffold surface compared with the bulk solution. Such lower local pH would provide more optimal pH for GOx or HRP. In another study, Xiong et al. [77] positioned GOx at different locations of a 3D octahedral DNA scaffold, providing different polyanionic environments for enzymes. By using the electrical sensing based on a bipolar junction transistor, the proton generations by enzyme at different locations were measured. The activity enhancement of DNA scaffold-tethered GOx was observed over the free GOx in the solution. In particular, GOx positioned at one vertex of the DNA octahedral exhibited a faster oxidation rate than when embedded between the bundles of the scaffold edge (Figure 6). This was explained by the different local pH environments induced by the DNA scaffold due to the exposure of GOx to different amounts of negatively charged DNA scaffold. The local pH change may be one of the reasons resulting in the catalytic enhancement, but the enhancement effect should be limited to particular enzymes, such as GOx or HRP that exhibits similar pH dependence with an increased maximal turnover rate under more acidic conditions. Therefore, the question of whether the local pH environment of DNA scaffold could modulate the reaction of other enzymes with various optimal pH preferences was asked as described in Section 3.7.

### 3.7. General Factors for the Catalytic Enhancement of DNA-Scaffolded Enzymes

As reported previously [78], two enzymes with different pH preferences, xylose reductase (XR) and xylitol dehydrogenase (XDH), were individually assembled on the fully open state of a 3D DNA scaffold through the modular adaptor method [79,80,81,82,83,84] in high loading yields. XR was genetically fused to the modular adaptor ZF-SNAP to obtain ZS-XR. The zif268 bound with the specific DNA sequence, while SNAP-tag would react with benzylguanine incorporated in the DNA sequence to form the covalent linkage [80]. Similarly, XDH was fused to the C-terminal of modular adaptor Halo-GCN4 to construct enzyme HG-XDH. The Halo-tag substrate 5-chlorohexane (CH) was incorporated in the GCN4-binding DNA sequence. The catalytic enhancements were observed for both the DNA-scaffolded ZS-XR (sXR) and scaffolded HG-XDH (sXDH) over the respective free enzyme (Figure 7). In the enzyme reactions, XR converted xylose with the cofactor NADH to xylitol, while XDH produced xylulose from xylitol using NAD^+^. Such neutral or net negative charge of their substrates and cofactors indicated that the surface–substrate or –cofactor electrostatic attractive interaction could not account for the increase in activities of assembled enzymes. Instead, the large scaffold with high packing density of DNA helices improved the enzyme stability and reduced the adsorption of scaffolded enzymes to the reaction vessels, which could partly contribute to the catalytic enhancement of DNA-scaffolded XR or XDH. To assess the local pH environment of DNA scaffold, a dual-emission ratiometric pH indicator SNARF derivative was loaded on the DNA scaffold either facing near the surface or locating 6.7 nm away from the surface, which corresponded to the distance between the enzyme and the surface of the DNA scaffold. The local pH near the surface of the DNA scaffold or near the enzyme loaded position in the reaction buffer (pH 7.0) was deduced to be 6.2 or 6.5. Such local pH shifts would result in 25% enhancement of the catalytic activity for sXR and 30% reduction for sXDH since XR and XDH displayed the optimal pH at 6.0 and 8.0, respectively. Therefore, the postulated modulation of enzyme activity by the lower pH shift near the DNA scaffold surface unlikely explained the catalytic enhancements of both scaffolded enzymes [78]. 

This study evaluated the DNA scaffolding effect on enzyme reactions from various aspects. The protection of the enzyme against adsorption to the reaction vessels by a DNA scaffold to benefit the enzyme reactions would be one of the general factors for the catalytic enhancement. However, this factor alone could not account for the large enhancement. The question that remains unanswered is, what is the critical character of the DNA scaffold for accelerating the enzyme reaction? The high-density charge of the DNA scaffold surface exerts the formation of an ordered hydration layer. Such a local microenvironment has been suggested to affect the catalytic reactions [75]; however, the chemical mechanisms remain to be elucidated.

**Figure 7 molecules-27-06309-f007:**
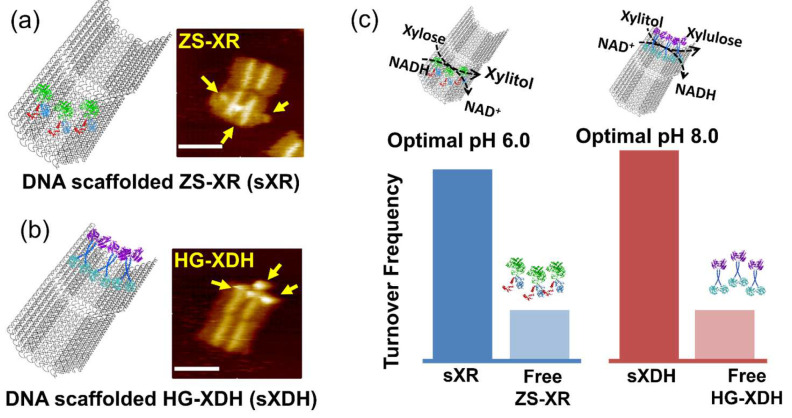
Xylose reductase (XR) and xylitol dehydrogenase (XDH) with different optimal reaction pH were individually assembled on the DNA scaffold [78]. (**a**) ZS-XR (modular adaptor ZF-SNAP fused XR) assembled on the DNA scaffold (sXR) and its AFM image, scale bar: 100 nm. The arrows in yellow indicated enzymes on the DNA scaffold. (**b**) HG-XDH (modular adaptor Halo-GCN4 fused XDH) assembled on the DNA scaffold (sXDH) and its AFM image, scale bar: 100 nm. (**c**) Turnover frequency of the enzyme reactions of free or scaffolded enzymes. Reprinted with permission from Ref. [78]. Copyright 2021 Royal Society of Chemistry.

### 3.8. Packed State of Enzymes on the DNA Scaffold

In cells, enzyme reactions were performed in highly packed conditions. To mimic this “cellular crowed environment” in vitro still remains challenging. Huyen et al. [85] applied the modular adaptor method to assemble the monomeric carbonic anhydrase (CA) on a DNA scaffold in the packed state or dispersed states. CA was genetically fused to the modular adaptor ZF-SNAP to obtain ZS-CA. In the packed state, the interenzyme distance between CA regions was less than 1 nm. The reactions of ZS-CA assembled on the DNA scaffold were performed with the substrate *p*-nitrophenyl acetate (*p*-NPA), *p*-nitrophenyl butyrate (*p*-NPB) or *p*-nitrophenyl valerate (*p*-NPV). Interestingly, the enzymatic reactions proceeded faster in the packed than in the dispersed state under same enzyme and substrate concentrations. Acceleration of the reactions in the packed assembly was more predominant for substrates with higher water-excluded volumes (higher hydrophobicity), in which the reactions were accelerated by 1.3-fold, 1.5-fold or 1.6-fold in the packed state over the dispersed state with *p*-NPA, *p*-NPB or *p*-NPV as the substrate (Figure 8). The entropic force of water increasing the local substrate concentration within the domain confined between enzyme surfaces was attributed to the acceleration of enzyme reactions in the packed assembly. The acceleration of the enzyme reaction in the packed state over the dispersed state was also observed for xylose reductase assembled on the same type of DNA scaffold. This system provides a reasonable molecular model of enzymes in a packed state inside the cell, such as the condensate in the liquid–liquid phase separation.

## 4. Enhanced Efficiency of Enzyme Cascade Reactions on the DNA Scaffold

The dependence of the cascade reaction efficiency on the interenzyme distance has been investigated extensively by using a variety of DNA scaffolds [17]. While the proximity effect of enzyme cascade reactions has been observed for several enzyme pairs, the actual mechanism for the observed interenzyme distance dependence of GOx/HRP cascade enzymes arouse much controversy [86]. As speculated for the confined cellular environment, the confined 3D DNA space was hypothesized to provide a more favorable environment for enzyme cascade reactions. In addition, the spatial arrangement of the enzyme and the balanced enzyme kinetics of the cascaded enzymes were also suggested to be the critical factors for the enzyme cascade reactions on the DNA scaffold [87]. In this section, we review the examples of enzyme cascade reaction on the DNA scaffold and discuss the previously proposed mechanisms for the enhanced cascade efficiency.

### 4.1. Activity Enhancement through the DNA Scaffold

The DNA scaffolding effect was also reported for the multienzyme on the DNA scaffold. William et al. [88] assembled a three-enzyme sequential cascade of amylase, maltase and glucokinase on a DNA origami triangle. The kinetics of seven different enzyme conjugations were evaluated experimentally and compared to the simulations of optimized activity. The initial catalytic rate enhancement factors were plotted for each cascade configuration as shown in Figure 9a. The overall 19-fold increase in the turnover rate for the 2:2:2 (amylase: maltase: glucokinase) configuration was not significantly higher than the 18-fold increase estimated from adding the individual enhancements for the 2:0:0, 0:2:0 and 0:0:2 configurations together. A 30-fold increased turnover rate was observed for the 2:4:2 enzyme cascade on the DNA scaffold. The detailed kinetic analysis suggested that this enhanced cascade efficiency was derived from the increased enzyme stability and the localized DNA surface affinity or hydration layer.

In another study, Lim et al. [89] described a versatile CRISPR-associated (Cas) nuclease-based strategy to construct multienzyme complexes scaffolded on a DNA tile (Figure 9b). Catalytically inactive Cas nuclease was used in combination with SpyCatcher-SpyTag conjugation strategy to assemble enzymes in programmable patterns. Five enzymes involved in the violacein biosynthesis pathway were precisely organized on the DNA scaffold in an “optimal” or “scrambled” pattern, in which the enzymes were arranged by the order of the sequential reactions in violacein biosynthesis as the “optimal” pattern. However, the effect derived from the enzyme arrangement patterns on the cascade efficiency was not significant. An increase (1.8-fold) in violacein production was observed for the reaction mixture consisting of five enzymes individually attached on the different DNA scaffolds over the free enzymes, demonstrating the benefits of the DNA scaffolding. A notable increase (3.2-fold) was observed for the five enzymes co-assembled on the DNA template compared with the free enzymes. This enhancement of cascade efficiency was attributed to two primary effects: activation effect from the DNA scaffold and close proximity of enzymes.

**Figure 9 molecules-27-06309-f009:**
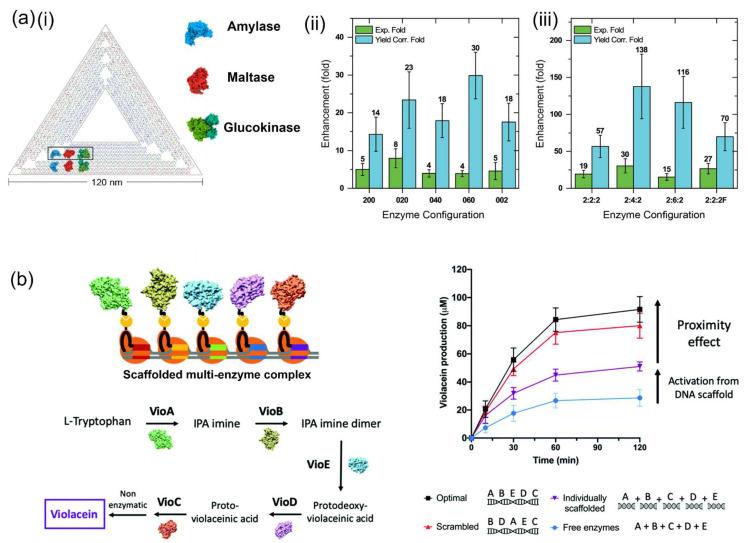
Activation of enzyme cascade reactions on the DNA scaffold. (**a**) Amylase, maltase and glucokinase co-assembled on a triangular DNA scaffold. (**i**) A scheme presenting the co-assembly of amylase, maltase and glucokinase on the DNA scaffold. (**ii**) Fold enhancement for the individual enzyme configurations. The green bars represent the experimentally determined enhancements (Exp. Fold). The blue bars represent the theoretical max enhancement obtainable assuming 100% attachment yield (Yield Corr. Fold). (**iii**) Fold enhancements for the full enzyme cascade configurations [88]. (**b**) Scheme of the CRISPR/Cas-directed programmable assembly of five enzymes on a DNA scaffold to convert L-tryptophan into violacein [89]. Reprinted with permission from: (**a**) Ref. [88]. Copyright 2019 American Chemical Society; (**b**) Ref. [89]. Copyright 2020 Royal Society of Chemistry.

### 4.2. Swinging Arms Facilitating the Substrate Channeling

To mimic the substrate channeling in natural systems, swinging arms modified with the cofactor were utilized to promote the enzyme cascade reactions on the DNA scaffold. The two-step reaction consisting of glucose-6-phosphate dehydrogenase (G6pDH) and malic dehydrogenase (MDH) was conducted on a DNA double-crossover (DX) tile scaffold with NAD^+^-tethered poly(T)_20_ oligonucleotide facilitating the transportation of NADH (Figure 10a) [90]. This NAD^+^-modified arm was positioned on 7 nm from either enzyme in a parallel geometry. For the same total NAD^+^ and enzyme concentrations, the overall activity of the G6pDH/MDH assembled on DNA scaffold with NAD^+^-modified arm was 90-fold higher than that of a two-enzyme complex co-assembled on DNA scaffold but with freely diffusing NAD^+^. In another study, G6pDH and lactic dehydrogenase (LDH) were co-assembled on a wireframe DNA origami scaffold with the swinging arms carrying NAD^+^ to facilitate the transport of the redox intermediate of NAD^+^/NADH between the cascaded enzymes (Figure 10b) [91].

To regulate the multiple enzyme cascade reactions, a four-arm Holliday junction with attached NAD^+^ was utilized as the transporting arm for the enzymes G6pDH, MDH and LDH arranged on a DNA origami scaffold [92]. By applying the toehold-mediated strand displacement mechanism, the NAD^+^-modified swinging arm was able to be switched from anchor 1 to anchor 2 to activate or block the enzyme pathway. The NAD^+^ substrate channeling in the enzyme pair G6pDH/MDH was promoted when the Holliday junction bound to anchor 1, resulting in the activation of G6pDH/MDH cascade reaction. By contrast, the enzyme pathway of G6pDH/MDH was blocked when the Holliday junction was switched to anchor 2, which activated the cascade reaction of the enzyme pair G6pDH/LDH (Figure 10c). Such switch of swinging arm can also be triggered by the external stimuli such as light instead of using DNA strand displacement [93].

**Figure 10 molecules-27-06309-f010:**
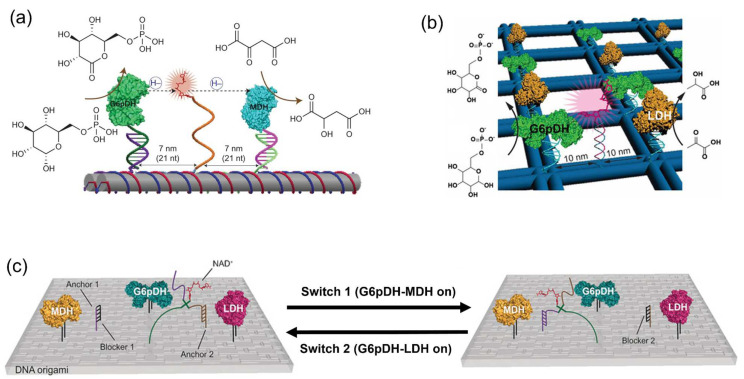
Enzyme cascade reactions on the DNA scaffolds with swinging arm. (**a**) Glucose-6-phosphate dehydrogenase/malic dehydrogenase (G6pDH/MDH) assembled on the DNA scaffold with the swinging arm modified with NAD^+^. The NAD^+^-modified single-stranded poly(T)_20_ is positioned halfway between the two enzymes, facilitating the transfer of hydrides [90]. (**b**) Glucose-6-phosphate dehydrogenase/lactic dehydrogenase (G6pDH/LDH) assembled on the DNA scaffold with the swinging arm modified with NAD^+^. The double-stranded DNA swinging arm carrying an NAD^+^ cofactor at the tip was positioned halfway between each pair of G6pDH and LDH to transfer the hydride intermediate [91]. (**c**) G6pDH, MDH and LDH co-assembled on the DNA scaffold with a NAD^+^-anchored Holliday junction. In the presence of switch 1, the NAD^+^ substrate channeling binds to anchor 1 and is located between G6pDH and MDH, thus driving the G6pDH/MDH pathway. In the presence of switch 2, the substrate channeling is located between G6pDH and LDH, driving the G6pDH/LDH pathway [92]. Reprinted with permission from: (**a**) Ref. [90]. Copyright 2014 Springer Nature; (**b**) Ref. [91]. Copyright 2018 John Wiley and Sons; (**c**) Ref. [92]. Copyright 2016 John Wiley and Sons.

### 4.3. Proximity Effect: Controversy Remains

Artificial enzyme cascades have been implemented on 2D DNA scaffolds with the study of proximity effect. In particular, the enzyme pair GOx/HRP was widely used due to the commercial availability and the stability of enzymes. Wilner et al. [94] activated the GOx/HRP enzyme cascade on the topologically programmed DNA scaffolds. The efficiency of GOx/HRP on a two-hexagon scaffold with the interenzyme distance of approx. 6 nm was 1.2-fold higher than that on a four-hexagon system (approx. 23 nm) (Figure 11a). This phenomenon was attributed to the close interenzyme distance that facilitated the transportation of intermediate H_2_O_2_. Fu et al. [95] organized GOx/HRP on a rectangular DNA scaffold by changing the interenzyme distance from 10 nm to 65 nm (Figure 11b). A drastic enhancement of activity at a distance of 10 nm was observed. It was suggested that Brownian diffusion of intermediates in solution would govern the cascade efficiency of GOx/HRP with far interenzyme distance (20–65 nm). In addition, the strong activation of the cascade reaction with a 10 nm interenzyme spacing was contributed by the dimensionally limited diffusion of intermediates crossing the closed protein. 

Besides the static DNA scaffolds, the simple dynamic DNA structures were also applied to investigate the enzyme cascade reaction of GOx/HRP. As a typical example, a DNA tweezer built by Xin et al. [96] could switch between open and closed states driven by a DNA strand displacement reaction (Figure 11c). The observed higher overall enzyme reaction efficiency of GOx/HRP in the closed state was attributed to the shorter distance of two enzyme-modified DNA arms that promoted the intermediate channeling. 

The results above support the notion that interenzyme distance is one of the important factors to accelerate the enzyme cascade reaction. However, controversy remains whether the interenzyme distance indeed affects the cascade efficiency of GOx/HRP. The proximity effect was suggested to occur only under conditions of limited time (very initial reaction state) [97], low concentration of intermediates [98] or in the presence of competing enzymes [99]. On the other hand, Zhang et al. [76] have proposed that catalytic enhancement of GOx/HRP resulted from the local pH decrease modulated by the negatively charged DNA scaffold surface rather than the close proximity of GOx and HRP.

Huyen et al. [100] assembled a RuBisCO dimer a DNA scaffold using a dimeric DNA binding protein as an adaptor. To mimic the environment of the natural microcompartment in cyanobacterial carboxysome, RuBisCO was co-assembled with carbonic anhydrase (CA) on the DNA scaffold (Figure 11d). While the natural carboxysome assembly is believed to enhance the RuBisCO activity by the preferential CO_2_ dehydration by CA, the co-assembled system showed a slower RuBisCO reaction rate. A similar observation was reported for the co-assembled system of RuBisCO and CA in a protein cage [48]. These results suggest that the proximity in the interenzyme distance of RuBisCO and CA is not the crucial determinant for the enhanced RuBisCO activity in carboxysome.

**Figure 11 molecules-27-06309-f011:**
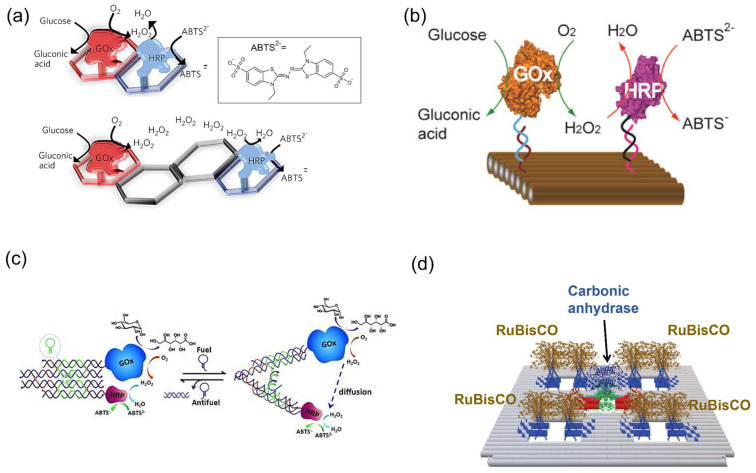
Enzyme cascade reactions on the DNA scaffolds to study the enzyme proximity effect. (**a**) Assembly of the GOx and HRP enzymes on the two-hexagon or four-hexagon strips [94]. (**b**) GOx/HRP assembled on the rectangular DNA scaffold [95]. (**c**) Enzyme cascade reaction of GOx and HRP was regulated by a dynamic DNA tweezer. The DNA machine contains DNA double crossover (DX) motifs constructing two rigid arms jointed by an immobile four-way junction [96]. (**d**) RuBisCO and carbonic anhydrase co-assembled on a 2D DNA scaffold [100]. Reprinted with permission from: (**a**) Ref. [94]. Copyright 2009 Springer Nature; (**b**) Ref. [95]. Copyright 2012 American Chemical Society; (**c**) Ref. [96]. Copyright 2013 John Wiley and Sons; (**d**) Ref. [100]. Copyright 2019 Elsevier Ltd.

### 4.4. Enzyme Kinetics of Cascade Reactions

In our previous studies, the enzyme cascade reactions derived from D-xylose metabolic pathway were investigated on 2D and 3D DNA scaffolds. The modular adaptor method was applied to stably co-locate enzymes on scaffold with the precise control over enzyme stoichiometry and interenzyme distance [79,80,81,82,83,84]. ZS-XR [80] and G-XDH (adaptor GCN4 fused xylitol dehydrogenase) [101] were arranged on a three-well DNA scaffold with the alteration of interenzyme distance from 98 nm to 10 nm; it was found that the highest efficiency appeared at a distance of 10 nm, and the intermediates (xylitol and NAD^+^) transferred to the downstream enzyme following the Brownian motion (Figure 12a) [80]. The enzyme cascade reaction XR/XDH was extended to realize the three-step reaction, in which xylulose kinase (XK) converted xylulose to xylulose-5-phosphate by utilizing ATP on the same 2D DNA scaffold [81]. The produced ADP in the system that co-assembled the third enzyme within 10 nm was higher than that with 50 nm. However, the distance dependence was observed to a smaller extent than that observed for the XR/XDH system, indicating that the kinetic parameters of upstream and downstream enzymes were the critical factor on the interenzyme distance dependence of cascade reaction efficiency (Figure 12b).

To maintain the same enzyme loading numbers with varying the interenzyme distance, a 3D DNA hexagonal prism (HP) with the shape transformation from the open state to the closed state triggered by DNA linkers was constructed to investigate an enzyme cascade reaction by XDH and XK [102]. In this study, the modular adaptor fused enzymes, HG-XDH (modular adaptor Halo-GCN4 fused XDH) and AC-XK (modular adaptor AZ-CLIP fused XK), were assembled on the open state of DNA scaffold in high enzyme assembly yield (HPO/3XDH-XK), followed by the addition of DNA linkers at 1:1 molar ratio to obtain the closed state of enzyme assembly (HPC/3XDH-XK) (Figure 12c). Time-course of the closing process of DNA scaffold assembled with enzymes was monitored by the Cy5 fluorescence intensity indicated over 90% closing yield after incubating for 12 h at 25 °C. The cascade reaction efficiency was analyzed by HPLC to quantitate the cofactors ATP, NAD^+^, NADH and ADP consumed or generated in the cascade reaction at steady-state conditions after 24 h reaction at 25 °C. The interenzyme distances of XDH-XK in open and closed states were estimated to be 60 nm and 20 nm, respectively. The interenzyme distance dependence was not significant for the cascade reaction of XDH-XK on the DNA scaffold, which is likely due to the fast kinetics and low *K*_m_ value of the second enzyme. The *K*_m_ values of HG-XDH and AC-XK for xylitol and xylulose are 104 mM and 0.18 mM, respectively. The *k*_cat_ values of HG-XDH and AC-XK are 59 min^−1^ and 12600 min^−1^, respectively. By comparing with the obvious interenzyme distance dependency observed for the XR/XDH cascade reaction reported previously in our laboratory [80], it was suggested that the balanced enzyme kinetic parameters played an important role in exerting the interenzyme distance dependency of cascade reaction efficiency.

**Figure 12 molecules-27-06309-f012:**
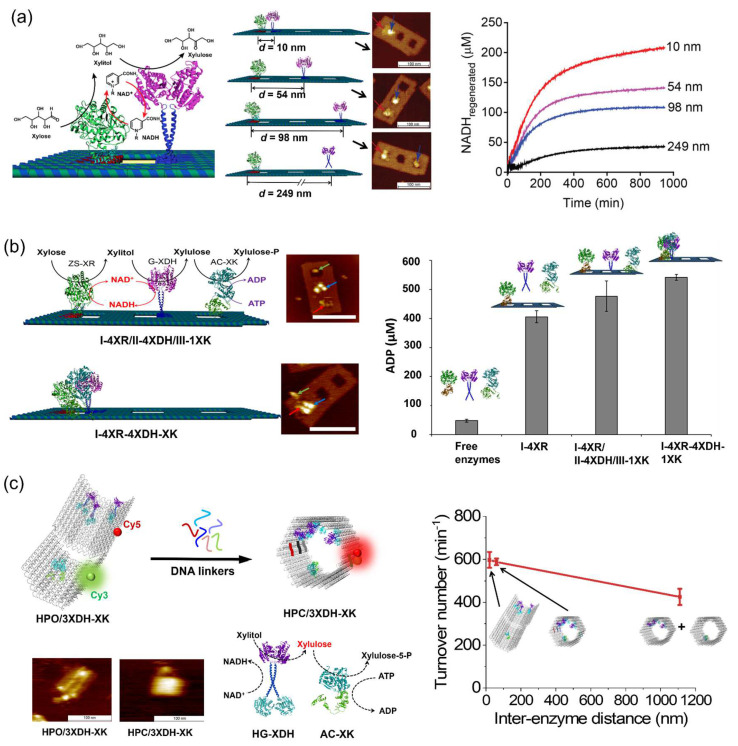
Enzyme cascade reactions derived from the D-xylose metabolic pathway studied on the 2D or 3D DNA scaffold. (**a**) Enzyme cascade reactions of xylose reductase/xylitol dehydrogenase (XR/XDH) on the DNA scaffold. By changing the enzyme locations of XR and XDH, the interenzyme distance was regulated [80]. (**b**) Xylose reductase/xylitol dehydrogenase/xylulose kinase (XR/XDH/XK) cascade reactions on the DNA scaffold. The ADP production used a measure of the cascade reaction efficiency [81]. (**c**) Enzyme cascade reactions of XDH and XK assembled on a 3D DNA hexagonal prim (HP) with the dynamic shape transformation. The dynamic transition from the open state of HP (HPO) to the closed state (HPC) was induced by the addition of DNA linkers that hybridized with the two domains of HP [102]. Reprinted with permission from: (**a**) Ref. [80]. Copyright 2016 American Chemical Society; (**b**) Ref. [81]. Copyright 2017 American Chemical Society; (**c**) Ref. [102]. Copyright 2021 Royal Society of Chemistry.

### 4.5. Spatial Arrangement of Enzymes

A three-enzyme cascade involving malic dehydrogenase (MDH), oxaloacetate decarboxylase (OAD) and lactate dehydrogenase (LDH) was located in a triangular pattern on a three-point star DNA scaffold with optimized geometric organization to enhance the production of lactate (Figure 13a) [103]. This study demonstrated that the overall activity of three-enzyme pathway relied less on the interenzyme distance and more on the geometric patterns with a 10 to 30 nm enzyme spacing. However, the benefits of the optimal arrangement of the enzyme on the cascade efficiency may be limited to the specific enzyme pairs. For the cascade reaction involved in the violacein pathway, the effect of enzyme arrangement patterns was not significant [89]. Compared with the 2D DNA scaffolds, the 3D DNA scaffolds provide more ideal space for the spatial arrangement of enzymes. Kahn et al. [104] created a library of 3D DNA wireframe octahedron structures for the spatial organization of GOx and HRP. The contribution of enzyme spacing, arrangement and location on the 3D DNA scaffold to cascade efficiency was investigated (Figure 13b). An increase in activity at small enzyme spacing of less than 10 nm between GOx and HRP was observed. Interestingly, the discontinuities in DNA scaffold structure was found to cause the activity changes of enzymes. 

The effect derived from the organization of enzymes on the scaffold may be affected by other factors such as enzyme spacing, enzyme kinetics or enzyme density. In a study by simulations, Chado et al. [105] evaluated the role of dimension and spatial arrangement on the activity of enzyme cascade reactions on the scaffolds. The enzyme cascade efficiency was weakly dependent on the spatial arrangement under the diffusion-limited conditions. At short length scales (i.e., sub 10 nm dimensions) and high enzyme densities, the effect of the spatial arrangement of enzymes on the cascade efficiency was insignificant.

**Figure 13 molecules-27-06309-f013:**
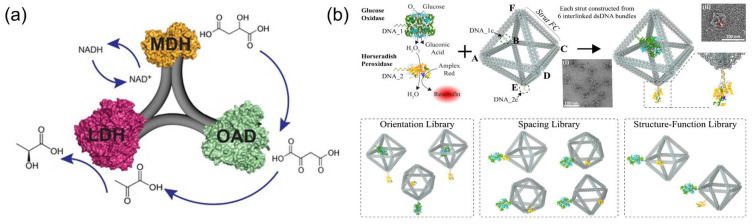
Enzyme cascade reactions on the DNA scaffolds to study the effect of spatial arrangement of enzymes. (**a**) Three types of enzyme, malic dehydrogenase/oxaloacetate decarboxylase/lactate dehydrogenase (LDH/MDH/OAD), co-assembled on a three-point star DNA scaffold to produce lactic acid from malic acid [103]. (**b**) The spatial arrangement of gOx and HRP on the DNA wireframe octahedrons [104]. Reprinted with permission from: (**a**) Ref. [103]. Copyright 2016 John Wiley and Sons.

### 4.6. Effect of 3D Confinement

As speculated for the confined cellular environments, a confined 3D DNA space could provide a favorable environment for enzyme cascade reactions [106]. The typical enzyme pair GOx/HRP was also investigated in the 3D DNA nanostructures. Fu et al. [107] observed the enhanced cascade efficiency of GOx/HRP encapsulated in a DNA nanotube compared with the free enzymes in bulk solution and suggested that local concentration of intermediate (H_2_O_2_) was increased due to the restricted diffusion in the confined DNA nanotube (Figure 14a). Linko et al. [108] constructed a DNA nanoreactor which was fabricated by gluing two separate origami units equipped with GOx and HRP (GOx-origami and HRP-origami) via the hybridization of 32 short bridging DNA sequences (Figure 14b). It was found that enzyme cascade reaction of GOx and HRP inside this DNA tube proceeded efficiently. In this study, the barrier effect restricting the diffusion of H_2_O_2_ derived from the DNA origami structure was proposed. In another study, by encapsulating GOx/HRP in the DNA nanocage, Zhao et al. [75] hypothesized that the ordered hydration layer near the negatively charged DNA cage increased enzyme stability and enhanced the cascade reaction efficiency (Figure 14c). While the factors such as the higher local concentration of intermediate derived from the restricted diffusion and the ordered hydration layer have been proposed for the elucidation of the DNA confined environment, the actual chemistry behind it remains elusive.

## 5. Conclusions and Future Perspectives

DNA origami has emerged as an outstanding way to create versatile predesigned 2D or 3D DNA structures, providing excellent platforms for the assembly of various biomolecules. The spatial arrangement of enzymes on the DNA scaffold with the controllable interenzyme distance and proper stoichiometry opens up promising ways to explore the working mechanisms of enzymatic reactions in the cell. To attain a better understanding, we summarized the proposed activation mechanisms derived from the characteristics of DNA scaffold and the factors regulating the scaffolded cascade reactions (Figure 15).

The accelerated reactions of single type of enzyme on the DNA scaffold have been widely observed. The previously proposed mechanisms, such as the substrate affinity to the negatively charged DNA scaffold by electrostatic interaction or the lower local pH on the scaffold surface, are not the general factors for the catalytic enhancement. The protection of the DNA-scaffolded enzyme against deactivation could be a general factor to enhance the activity of the scaffolded enzyme. In particular, the reduced adsorption of scaffolded enzymes on the reaction vessels is certainly a mechanism to maintain the catalytic ability of enzymes, but this effect alone would not account for the widely observed catalytic enhancement of DNA-scaffolded enzymes. While the ordered hydration layer attracted by the negatively charged DNA scaffold surface is proposed to be an important factor for the catalytic enhancement by stabilizing the scaffolded enzyme, the chemistry behind this hypothesis has not yet been elucidated. The actual mechanisms of the DNA scaffold modulating the enzymatic reactions remain to be further studied.

The mechanism of DNA scaffolding effect for the single type of enzyme can also be applied for the multienzyme on the DNA scaffold. Besides this, the enhanced efficiency of enzyme cascade reaction on the DNA scaffold has been attributed to the close proximity of enzymes. From the kinetic study, it was found that the proximity effect does not contribute to the efficiency of enzyme cascade with imbalanced enzyme kinetics. The confined environment of the DNA scaffold has been suggested to increase the local concentration of substrate and enhance the enzyme cascade reaction. However, again, the chemistry behind this favorable “DNA confinement” remains elusive. Compared with the 2D DNA scaffold, the 3D DNA scaffold displays greater potential for the spatial arrangement of enzymes in various patterns, which would facilitate the investigation of enzyme cascade reaction from multiple factors, such as spacing, orientation and stoichiometry of enzymes.

At present, most of the artificial enzyme cascades on the DNA scaffold were designed as static models for the enzymes, representing the snap-shot of the biochemical processes implemented in cells. While the static models are useful for characterizing each factor to delineate its role on the efficiency of cascade enzyme assembly, an intelligent system that couples the dynamic shape transformation of the scaffold with assembly and disassembly by the sequential enzymes enables further investigation on the dynamic output of the enzyme cascades. Understanding the mechanistic aspects of the DNA scaffolding effect on the enzymatic reactions and the regulation factors for the enzyme cascade reactions on the DNA scaffold will provide new strategies for the construction of efficient biocatalysis and realize more complex artificial metabolic systems in vitro not only limited to the DNA scaffold but also for other artificial scaffolding systems.

**Figure 15 molecules-27-06309-f015:**
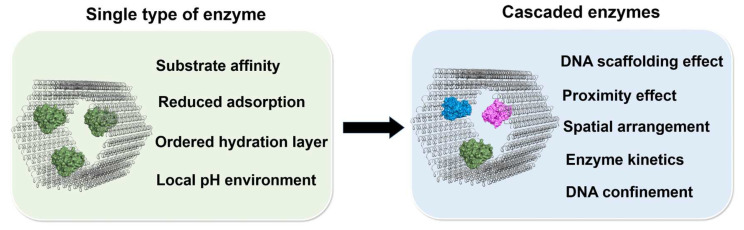
Mechanisms of the catalytic enhancement of single type of enzyme (either single or multiple in number) assembled on the DNA scaffold or the enhanced efficiency of enzyme cascade reactions on the DNA scaffold.

## Figures and Tables

**Figure 4 molecules-27-06309-f004:**
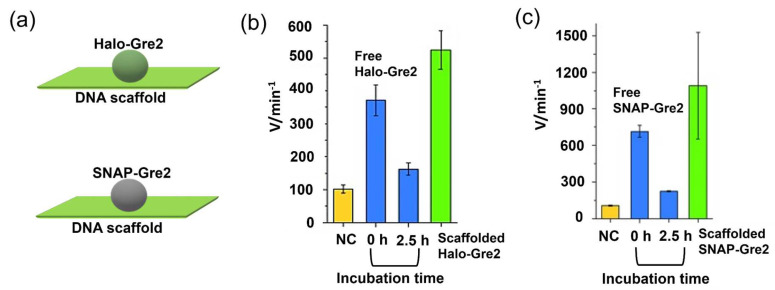
The *S*-selective NADP^+^/NADPH-dependent oxidoreductase Gre2 was assembled on the DNA scaffold with the study of protection effect of scaffold [71]. (**a**) Enzyme Gre2 was fused to Halo-tag or SNAP-tag to obtain Halo-Gre2 or SNAP-Gre2, which was then assembled on the DNA scaffold. (**b**) Comparison of reaction velocities estimated for DNA scaffolded Halo-Gre2 (green bar) as well as free Halo-Gre2 (blue bars), freshly prepared (0 h) or incubated for 2.5 h. NC (negative control) indicated the sample containing buffer only. (**c**) Comparison of reaction velocities estimated for DNA scaffolded SNAP-Gre2 (green bar) as well as free SNAP-Gre2 (blue bars), freshly prepared (0 h) or incubated for 2.5 h. NC (negative control) indicated the sample containing buffer only. Reprinted with permission from Ref. [71]. Copyright 2015 John Wiley and Sons.

**Figure 5 molecules-27-06309-f005:**
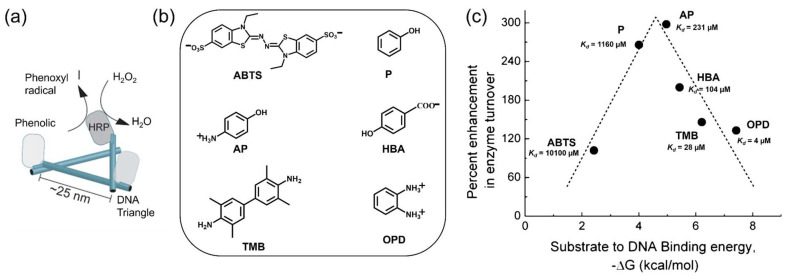
Horseradish peroxidase (HRP) was assembled on the DNA triangle with the study of substrate affinity to the scaffold [73]. (**a**) A scheme representing the assembly of HRP on the triangular DNA scaffold. (**b**) The structures of HRP substrates. 2,2′-azino-bis(3-ethylbenzothiazoline-6-sulphonic acid) (ABTS), phenol (P), *p*-Aminophenol (AP), *p*-hydroxybenzoic acid (HBA), 3,3′,5,5′-tetramethylbenzidine (TMB), *o*-phenylenediamine (OPD) were used in this study. (**c**) The enhancement of HRP activity on the DNA scaffold as a function of the predicted substrate–DNA binding energy. Reprinted with permission from Ref. [73]. Copyright 2013 American Chemical Society.

**Figure 6 molecules-27-06309-f006:**
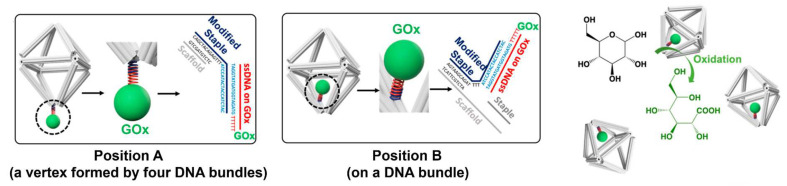
Glucose oxidase (GOx) was anchored on the different positions of the octahedral DNA scaffold to study the local pH environment of scaffold [77]. Reprinted with permission from Ref. [77]. Copyright 2020 American Chemical Society.

**Figure 8 molecules-27-06309-f008:**
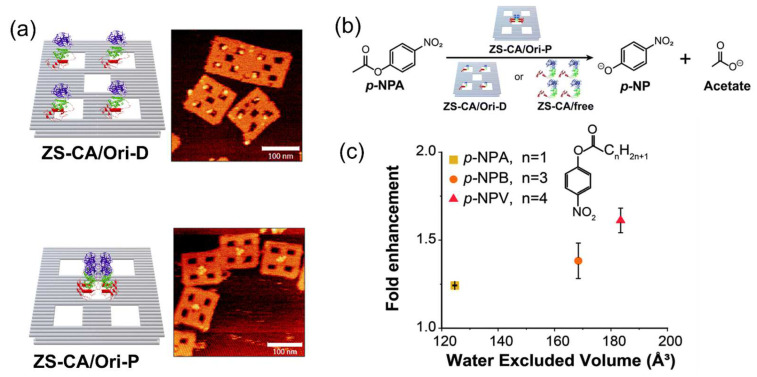
Carbonic anhydrase (CA) was assembled on the DNA scaffold in a dispersed or packed state [85]. (**a**) ZS-CA (modular adaptor ZF-SNAP fused CA) assembled on the DNA scaffold in a dispersed state (ZS-CA/Ori-D) or packed state (ZS-CA/Ori-P). The scale bar of the AFM images was 100 nm. (**b**) The reaction scheme presenting the hydrolysis of *p*-nitrophenyl acetate (*p*-NPA) catalyzed by CA. (**c**) The enhancement factors (initial velocity of CA on Ori-P over that of CA on Ori-D) for the esterase reaction of ZS-CA plotted against the water-excluded volume of *p*-NPA, *p*-nitrophenyl butyrate (*p*-NPB) and *p*-nitrophenyl valerate (*p*-NPV). Printed with permission from Ref. [85], Copyright 2021 Royal Society of Chemistry.

**Figure 14 molecules-27-06309-f014:**
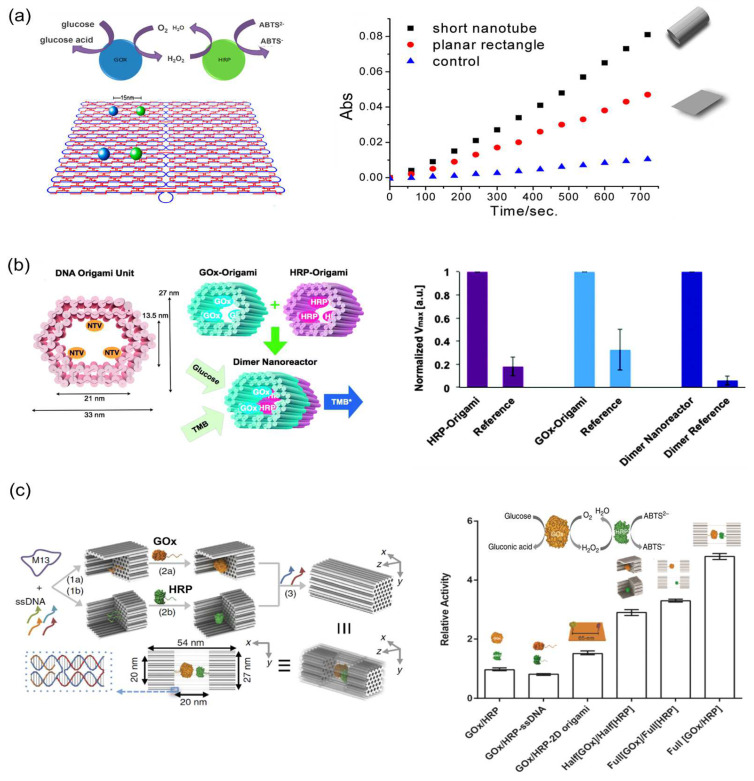
Enzyme cascade reactions on the DNA scaffolds with confined environment. (**a**) Illustrations of glucose oxidase (GOx) and horseradish peroxidase (HRP) assembled on the DNA rectangle or encapsulated in the DNA tube [107]. (**b**) GOx and HRP encapsulated in a DNA tubular structure. Two separately fabricated origami units are equipped with biotinylated GOx or HRP through biotin–avidin interaction. The right panel showed the maximum rate of reactions (*V*_max_) of enzymes attached to DNA origami units and for the assembled dimer nanoreactor. Dimer reference indicated the dimer origami fabricated similarly but without enzyme binding sites [108]. (**c**) GOx/HRP encapsulated in a DNA nanocage. Individual enzymes were first attached to half cages, followed by the addition of linker strands to combine the two halves into a full cage. The right panel showed the relative activity of a GOx/HRP pair when attached to a variety of DNA scaffolds: enzyme wild types (GOx/HRP), ssDNA (GOx/HRP-ssDNA), 2D rectangular DNA origami (GOx/HRP-2D origami), separate 3D half cages (Half[GOx]/Half[HRP]), separately full cages (Full[GOx]/Full[HRP]) and the same full cage (Full [GOx/HRP]) [75]. Reprinted with permission from: (**a**) Ref. [107]. Copyright 2013 American Chemical Society; (**b**) Ref. [108]. Copyright 2015 Royal Society of Chemistry; (**c**) Ref. [75]. Copyright 2016 Springer Nature.

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
