# Peer review of "Mechanistic Aspects for the Modulation of Enzyme Reactions on the DNA Scaffold"

_molecules, 2022, doi:10.3390/molecules27196309_

Round 1

Reviewer 1 Report

This review describes the use of DNA nanostructures for spatial organization of enzymes. Because the authors actively work in this field, the contents are comprehensive and very informative. I strongly recommend its publication.

However, this excellent manuscript will be much improved by the following changes, and increase the scope of its readability. My primary concern is that this review has been written primarily for the people who are working in this field. For them, this review should be very useful. But other readers cannot understand the contents well because of the following reasons.

1.     The legends for Figures (and Figures themselves) are not very informative, and should be changed. It would be more helpful if the authors present, in place of the Figures from the original references, the Figures which are made to show only the contents the authors should like to claim in this review.

For example, the legend for Figure 4 is as follows: “Single type of enzymes assembled on the DNA scaffolds. (a) Oxidoreductase Gre2 assembled on rectangular DNA scaffold [71]. (b) Left: assembly of HRP on the triangular DNA scaffold; right: the enhancement of HRP activity as a function of the predicted substrate-DNA binding energy [73]. (c) The kcat enhancement of different enzymes encapsulated in DNA nanocage [75]. (d) Glucose oxidase (GOx) positioned on the different positions of the octahedral DNA scaffold [77]”. In my opinion, this legend does not well explain the contents of Figure 4. In the right of Figure 4(a), for example, NC, 0 h, 2.5 h, and F36 are used without definition. In (c), the authors do not understand why MW is used as a parameter for the analysis of the results. In (d), what are optical sensor and BJT sensor? Under these conditions, the readers should wonder what they should understand from this Figure (direct reading of the original papers should be too much). If the authors should like to explain all of them to the readers, all these terms should be clearly described (at least in the text). On the other hand, if the purpose of Figure 4 is simply to present the examples of “Single type of enzymes assembled on the DNA scaffolds”, other parts (graphs) should be deleted, in my opinion.

Similarly, Figure 5 can be changed. It is difficult to understand the contents from this Figure alone. What are ZS-XR, free ZS-XR, and HG-XDH? What do arrows show? In many of the other Figures, I can point out similar things. It should be very kind to the readers, if they can have some idea simply by reading this review. Only when necessary, they can refer to the original papers.

2.     The title can be slightly changed? I am afraid that few readers can correctly understand the meaning of “scaffolded enzyme activity on DNA scaffold” when they see only the title.

Minor points: Too many abbreviations are used in the text. It is more kind, especially for the readers from different fields, if the number is reduced (or the meaning is shown appropriately). 

Author Response

Please see the attached pdf file.

Reviewer 2 Report

The authors summarized several enzymatic reactions on DNA scaffold and described the possible mechanisms of catalytic scaffolded enzymes under different conditions. They also discussed the controversy of artificial enzyme cascades on 2D DNA scaffolds and proposed possible future perspectives including multi-enzyme reaction on DNA scaffolds and on other artificial scaffolding systems. The review was well organized and was publishable after some minor revisions of typos and grammars.  For example, “Hela cells” should be “HeLa cells”;  the grammar should be consistent in this sentence, ”It was found purinosome comprises nine enzymes that act synergistically” et al….

Author Response

The authors summarized several enzymatic reactions on DNA scaffold and described the possible mechanisms of catalytic scaffolded enzymes under different conditions. They also discussed the controversy of artificial enzyme cascades on 2D DNA scaffolds and proposed possible future perspectives including multi-enzyme reaction on DNA scaffolds and on other artificial scaffolding systems. The review was well organized and was publishable after some minor revisions of typos and grammars.  For example, “Hela cells” should be “HeLa cells”;  the grammar should be consistent in this sentence, ”It was found purinosome comprises nine enzymes that act synergistically” et al….

We thank the reviewer’s comments. According to the suggestion, typos and grammars have been revised. The details were shown in the revised manuscript.